# Association between parental psychiatric conditions and offspring psychiatric, behavioral, and psychosocial outcomes: A Swedish population-based children-of-monozygotic twins study

Mengping Zhou[1]*, Henrik Larsson[1,2], Brian M. D'Onofrio[1,3], Mikael Landén[1,4], Ralf Kuja-Halkola[1], Zheng Chang[1], Isabell Brikell[1,5,6], Paul Lichtenstein[1], Erik Pettersson[1]

1 Department of Medical Epidemiology and Biostatistics, Karolinska Institutet, Stockholm, Sweden, 2 School of Medical Sciences, Örebro University, Örebro, Sweden, 3 Department of Psychological and Brain Sciences, Indiana University, Bloomington, Indiana, United States of America, 4 Institute of Neuroscience and Physiology, The Sahlgrenska Academy at Gothenburg University, Gothenburg, Sweden, 5 Department of Global Public Health and Primary Care, University of Bergen, Bergen, Norway, 6 Department of Biomedicine, Aarhus University, Aarhus, Denmark

* mengping.zhou@ki.se

## Abstract

### Background

Mental health problems tend to run in families, with studies showing transdiagnostic associations across generations. Nevertheless, if these associations were attributable to unmeasured familial (either environmental or genetic) factors that influence both generations, then treating the parental conditions would not break the intergenerational transmission. This study aims to investigate the associations between parental psychiatric conditions and offspring psychiatric, behavioral, and psychosocial outcomes, after controlling for unmeasured familial factors shared by offspring of monozygotic (MZ) twin parents (i.e., cousins).

### Methods and findings

We conducted a children-of-MZ twins study that consisted of 15,603 individuals (born to 7,742 MZ twin parents) born in Sweden between 1970 and 2000, and followed them from their date of birth to the date of the outcome or December 31, 2020, when the offspring were between 21 and 51 years old. The exposures were whether the MZ twin parents were diagnosed with any psychiatric condition, any internalizing condition, or any externalizing condition. The outcomes included register-based psychiatric conditions, behavioral problems, suicide, and psychosocial problems in the offspring. We performed stratified Cox regression for time-to-event outcomes and

**Data availability statement:** The Public Access to Information and Secrecy Act in Sweden prohibits us from making individual-level data publicly available. Researchers can apply for individual-level data through Statistics Sweden at: https://www.scb.se/en/services/guidancefor-researchers-and-universities/.

**Funding:** E.P. was supported by the Swedish Research Council (NO. 2017-01358; 2023-01999), Swedish Research Council for Health, Working Life and Welfare (2023-00402), Svenska Läkaresällskapet (SLS-943288), Stiftelsen Söderström-Königska (SLS-968742), and the Åke Wiberg Foundation. The funding sources had no role in the design and conduct of the study; the collection, management, analysis and interpretation of data; the preparation, review and approval of the manuscript; and the decision to submit the manuscript for publication.

**Competing interests:** I have read the journal's policy and the authors of this manuscript have the following competing interests: H.L. reports receiving grants from Shire Pharmaceuticals; personal fees from and serving as a speaker for Medice, Shire/Takeda Pharmaceuticals and Evolan Pharma AB; all outside the submitted work. H.L. is editor-in-chief of JCPP Advances. M.L. has received lecture honoraria from Lundbeck pharmaceuticals. Z.C. received lecture honoraria from Takeda Pharmaceuticals, outside the submitted work. All other authors have declared that they have no competing or potential conflicts of interest.

**Abbreviations :** ADHD, Attention-Deficit/Hyperactivity Disorder; CI, confidence interval; COT, children-of-twins; HRs, hazard ratios; ICD, International Classification of Diseases; MZ, monozygotic; OCD, obsessive-compulsive disorder; ODD, oppositional defiant disorder; OR, odds ratio; PPV, positive predictive value; PTSD, posttraumatic stress disorder; STROBE, Strengthening the Reporting of Observational Studies in Epidemiology.

conditional logistic regression for binary outcomes to compare offspring exposed to an MZ twin parent with psychiatric conditions against their unexposed cousins. We adjusted for the highest parental educational level, maternal and paternal age at childbirth, offspring birth year, offspring sex, and psychiatric diagnosis of the nontwin parent. Offspring of parents with any parental psychiatric condition, internalizing condition, or externalizing condition had significantly higher probabilities for all the psychiatric, behavioral, and psychosocial outcomes, with hazard ratios (HRs) ranging from 1.34 (95% confidence interval [CI] [1.21, 1.49]; $p < 0.001$) to 2.53 (95% CI [1.96, 3.26]; $p < 0.001$) for time-to-event outcomes and odds ratios ranging from 1.33 (95% CI [1.17, 1.52]; $p < 0.001$) to 1.87 (95% CI [1.63, 2.14]; $p < 0.001$) for binary outcomes. Although these associations attenuated when comparing differentially exposed cousins whose parents were MZ twins (20 out of 27 associations were no longer statistically significant within cousin pairs), associations between broad spectra remained statistically significant. Specifically, across the main analysis and several sensitivity analyses, statistically significant within-twin-family associations remained between any parental psychiatric condition and any offspring psychiatric condition (HR = 1.28, 95% CI [1.13, 1.44]; $p < 0.001$), between parental internalizing conditions and any offspring psychiatric condition (HR = 1.26, 95% CI [1.09, 1.45]; $p = 0.002$), and between parental externalizing conditions and any offspring psychiatric condition (HR = 1.27, 95% CI [1.08, 1.51]; $p = 0.005$). The main limitations of this study were unmeasured confounders not shared by cousins, the lack of diagnostic data from primary care, and limited statistical power for some specific clustered outcomes.

## Conclusions

Although the intergenerational transmission between parental psychiatric conditions and offspring psychiatric, behavioral, and psychosocial outcomes appeared partially attributable to unmeasured familial (environmental or genetic) factors that influenced both generations, there was also evidence of either nonshared factors or direct causal effects. If the latter, then treating parental psychiatric conditions would reduce the risk of psychiatric vulnerability in offspring.

## Author summary

### Why was this study done?

- The transdiagnostic nature of intergenerational mental health transmission is well-established, but it remains unclear if the associations might be attributable to unmeasured shared familial factors.

- Prior studies are limited by small sample sizes, narrow focus on a select few offspring psychiatric outcomes, and insufficient examination of broader psychosocial functioning and behavioral problems.

## What did the researchers do and find?

- We used the largest children-of-monozygotic- (MZ) twins sample to date, identifying 15,603 individuals born to 7,742 MZ twin parents in Sweden, followed until the date of outcome or December 31, 2020 (ages 21–51).

- When comparing differentially exposed cousins whose parents were MZ twins (i.e., controlling for familial factors), the associations attenuated substantially, and most (20/27) were no longer statistically significant.

- Nevertheless, the association of any parental psychiatric condition with any offspring psychiatric condition remained significant in the main and several sensitivity analyses.

## What do these findings mean?

- Familial factors (e.g., genetics, shared environments) likely contribute substantially to the intergenerational transmission of parental psychiatric conditions with offspring psychiatric, behavioral, and psychosocial outcomes.

- Nevertheless, as the associations between broad psychiatric spectra remained significant within cousin pairs, treating parental psychiatric conditions might partially reduce psychiatric vulnerability in offspring.

- The children-of-MZ-twins design cannot control for unmeasured factors not shared by cousins, meaning residual confounding factors could still be present.

- This study is based on clinical diagnoses assigned in specialist care, and therefore likely not generalizable to cases treated exclusively in primary care, or to individuals who did not seek treatment.

## Introduction

Mental health problems run in families, and both register-based and survey studies indicate that much of this intergenerational transmission appears transdiagnostic [1,2]. For instance, in a recent meta-analysis of 457 studies covering 3 million families, each of 10 psychiatric diagnoses in parents was associated with all 10 diagnoses in the offspring [3]. Additionally, this transdiagnostic transmission extends to nonpsychiatric outcomes, including suicide, poor school performance, and long-term unemployment in the offspring [4].

Mental health problems could run in families through multiple pathways. On the one hand, the parental psychiatric condition could create an adverse rearing environment that causally contributes to mental health problems in offspring [5,6]. If so, treating the parental condition would reduce the risk of mental health problems in the offspring. On the other hand, as parents provide not only the rearing environment but also 50% of their genes to their children, genetic factors might also explain the observed parent–offspring associations. That is, children whose biological parent suffered from a mental health condition might develop a mental health condition themselves, even if they had grown up with adoptive parents [7]. Likewise, any environmental factor shared by parents and their offspring (e.g., neighborhoods, socioeconomic status, parent's own child-rearing conditions, etc.) could also contribute to the observed parent-offspring associations. If the intergenerational transmission were entirely attributable to familial (genetic or environmental) factors that influence both the parent and offspring, then treating the parental condition would have no bearing on the risk of the offspring.

One way to control for familial factors is the children-of-twins (COT) design, which analyzes associations within twin parents and their respective children (i.e., cousins) to generate a within-twin-family effect. By comparing offspring exposed to a twin parent with psychiatric conditions against their unexposed cousins (i.e., cousins whose parent does not have a psychiatric condition), this approach adjusts for familial factors shared by cousins (Fig 1). Furthermore, by exclusively relying on monozygotic (MZ) twin parents, this design rules out 100% of genetic factors from the twin parent as children of

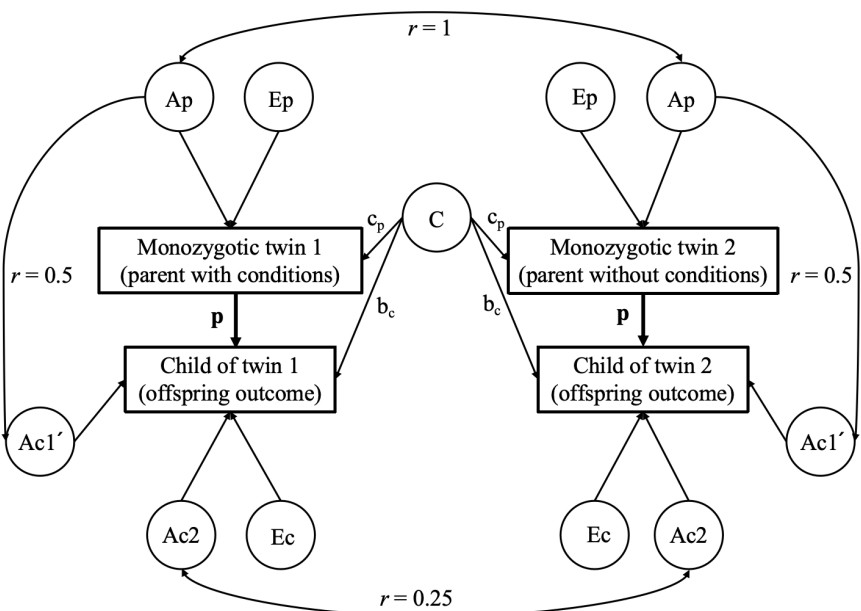

**Fig 1. Representation of the children-of-monozygotic-twins model.** Note: The phenotypic path labeled p describes the direct environmental effect of parental conditions on offspring outcomes. The children-of-twins model estimates p in observational data by comparing offspring exposed to a twin parent with psychiatric conditions against their unexposed cousins. This way, the model controls for the effect of unmeasured genetic (labeled $A_p$) and environmental factors (labeled C) shared by parents and offspring. Furthermore, as monozygotic twins share 100% of their genes, the offspring of MZ twin parents are as genetically to their own parent ($r = 0.5$) as they are to their uncle/aunt ($r = 0.5$), such that any observed differences between the cousins cannot be attributed to genetic differences in the parents. Ap = additive genetic effects on parent, Ep = nonshared environmental effects on parent, $c_p$ = shared environmental effects on parent, $b_c$ = effect of parental shared environmental factors on the child, Ac1′ = Additive genetic effects on child shared with the parent, Ac2 = Additive genetic effects on child, Ec = nonshared environmental effects on children, p = phenotypic effect of parent on child.

MZ twin parents are as genetically similar to their own parent as to that of their aunt or uncle (Fig 1) [6]. However, the COT design cannot control for unmeasured factors not shared by cousins.

Previous COT studies and those based on other genetically informative designs (e.g., adoption studies and sibling comparisons) have suggested that the intergenerational transmission of psychotic disorders appears entirely attributable to familial factors [5,8,9]. In contrast, evidence is mixed regarding the transmission of internalizing and externalizing conditions. For example, in a children-of-MZ twins study ($N_{offspring} = 445$) from the Australian Twin Register where the exposure was parent-reported depression diagnosis, and which additionally controlled for the observed depression status of the twin-parent's partner, found that the intergenerational transmission appeared consistent with a causal interpretation (or with the presence of nonshared unmeasured factors) when predicting offspring-reported depression (hazard ratio [HR] = 1.39, 95% confidence interval [CI] [1.00, 1.94]), but that the precision was too low to draw firm conclusions when predicting offspring-reported conduct disorder (odds ratio [OR] = 1.41, 95% CI [0.63, 3.14]) [10].

Two studies comparing larger samples of children of siblings ($N_{offspring} = 2,476,198$ and 44,250) based on Swedish national registers, which had more statistical precision and that relied on diagnoses assigned by psychiatrists in both the parent and offspring generation, reported a significant within-family intergenerational transmission of substance misuse [11,12]. However, this association might still be attributable to genetic factors, as the children-of-siblings design only adjusts for partial genetic relatedness. Accounting for more genetic factors, a systematic review of 12 COTs studies using the Australian Twin Registry ($N_{offspring} = 922–2,554$) and the Vietnam Era Twin Registry ($N_{offspring} = 831–1,917$) found no significant associations between parental substance misuse and offspring psychiatric outcomes [5], with the exception that a significant association was observed between parental alcohol/drug dependence and offspring conduct problems [13].

Although the above studies explored the within-family effect of parental internalizing and externalizing conditions on a select few offspring psychiatric outcomes, many psychiatric conditions, behavioral problems, and psychosocial outcomes in the offspring remain unexamined. In addition, much of past research on the intergenerational transmission of internalizing and externalizing conditions was based on retrospective parent- or self-reported symptoms, which can lead to recall or rater bias. Furthermore, past children-of-MZ twins studies were relatively small.

We aimed to examine if associations of any parental, internalizing, and externalizing conditions with offspring psychiatric conditions, externalizing behaviors, suicide, school performance, and unemployment in adulthood remained after controlling for unmeasured familial factors using the largest sample of children of MZ twins to date.

## Methods

### Study population

We identified all individuals born in Sweden between 1970 and 2000 from the Total Population Register and linked them to their parents and grandparents using the Multi-Generation Register and to several other national registers to obtain their psychiatric, violent crime, suicide, and psychosocial outcomes. Table A in S1 Appendix presents a detailed description of these registers.

We excluded individuals for whom we could not identify both biological parents and excluded stillbirths, children with congenital malformations, and those who died neonatally. We also excluded individuals with missing values on any covariates (as listed in 2.4 below). We identified children of MZ twins in two steps. First, we identified all cousins who shared the same maternal grandmother and grandfather or paternal grandmother and grandfather. Second, we identified the parents of the cousins who were identified as MZ twins via the Swedish Twin Register surveys, which have targeted nearly all twin pairs born in Sweden from 1886 through 2000 [14]. All individuals were followed from their date of birth to the date of the outcome, emigration from Sweden, death, or December 31, 2020, when the offspring were between 21 and 51 years old.

The Stockholm Regional Ethical Review Board approved this study (IRB approval number: Dnr 2020-06540), and informed consent was not required because it relied on registry data. This study is reported as per the Strengthening the Reporting of Observational Studies in Epidemiology (STROBE) guideline (S1 Checklist).

### Exposures

The exposures were parental diagnoses of any internalizing or externalizing conditions. We defined internalizing conditions based on whether parents had any lifetime diagnoses of anxiety, depression, posttraumatic stress disorder (PTSD), obsessive-compulsive disorder (OCD), or eating disorders. We defined externalizing conditions based on whether parents had any lifetime diagnoses of substance use disorders (i.e., alcohol-related disorders, drug-related disorders), oppositional defiant disorder (ODD), or a court conviction of a violent crime. We also included any psychiatric diagnosis as an exposure (International Classification of Diseases [ICD]-8: 290-319, ICD-9: 290-319, ICD-10: F00-99). We did not include parental psychotic and neurodevelopmental disorders as exposures for three reasons. First, these two exposures had limited statistical power due to low prevalences. Second, as discussed in the Introduction, evidence suggests that the intergenerational transmission of psychotic disorders appears almost entirely attributable to familial factors, making their inclusion less meaningful. Third, the primary neurodevelopmental diagnoses were not included when the inpatient registry began in 1973, which would lead to under-ascertainment of neurodevelopmental diagnoses among parents born before 1960 (corresponding to 66% of our sample), given their characteristic of early-life onset.

All diagnoses were identified according to the ICD; 8th (1969–1986), 9th (1987–1996), and 10th (1997-) revision [15,16]. The psychiatric diagnoses were assigned by specialists following in- (data available year 1973–2020) or outpatient (data available year 2001–2020) psychiatric contact.

## Outcomes

The limited number of available MZ twins raises issues about statistical precision, particularly when examining rare outcomes. To boost statistical power, we consolidated psychiatric, behavioral, and psychosocial outcomes into seven theoretically motivated clusters. The first cluster, any psychotic-like condition, consisted of records of schizophrenia, bipolar disorder, and prescription of antipsychotics or mood stabilizers (lithium and/or antiepileptics). The second cluster, any neurodevelopmental condition, consisted of records of Attention-Deficit/Hyperactivity Disorder (ADHD), tic disorder, autism spectrum disorder, intellectual disability, learning disorders, and prescription of stimulants. The third cluster, any internalizing condition, consisted of diagnostic records of anxiety, depression, PTSD, OCD, eating disorders, and prescription of anxiolytics or antidepressants. The fourth cluster, any substance use disorder, consisted of records of alcohol-related disorders, drug-related disorders, and prescription of anti-alcohol or anti-opioid medications. The fifth cluster, any externalizing behavior, referred to records of ODD or court convictions of violent crimes. The sixth cluster, suicide, included suicide attempts and death by suicide. The seventh cluster, poor school performance, was identified as ranking in the lowest quintile on the junior high school grade point average or attaining only a compulsory education level. In addition, as a separate variable, we included long-term (defined as at least one consecutive year) unemployment. Lastly, we also created an outcome cluster that included any psychiatric diagnosis (ICD-8: 290-319, ICD-9: 290-319, ICD-10: F00-99). Table B in S1 Appendix presents the ICD/ATC code, a description of convictions classified as violent crimes, and the minimum cutoff age (prior to which a diagnosis might be considered unreliable) for each exposure and outcome. Table A in S1 Appendix presents the corresponding registries from which these outcomes were extracted. Any psychiatric condition, clustered psychiatric outcomes, and suicide were treated as time-to-event variables, with the time-to-event defined as the earliest occurrence of any specific outcomes within each cluster. Poor school performance and long-term unemployment were treated as binary variables (i.e., ever recorded versus never recorded).

## Covariates

We included the highest parental educational level, maternal and paternal age at childbirth, offspring birth year, offspring sex, and psychiatric diagnosis of the nontwin parent (i.e., the other parent who was not an MZ twin) as measured covariates.

## Statistical analyses

We first compared offspring exposed to parental psychiatric conditions to offspring not exposed, generating between-family estimates. Second, we compared individuals to their cousins born to MZ twin parents (i.e., where one twin parent had a psychiatric condition, and the other twin parent did not), generating within-twin-family estimates. The between-family analyses compare all offspring exposed to parental psychiatric conditions against all unexposed offspring, thereby capturing (potential) direct causal effects, confounding from shared familial factors (e.g., genetics, environment, and socioeconomic status), and confounding from nonshared factors. In contrast, the within-twin-family analyses are conditional on family strata, thereby controlling for shared familial factors. To assess the between-family effects, we performed Cox regression for time-to-event outcomes and logistic regression for binary outcomes. To estimate the within-twin-family effects (i.e., fixed-effects or cousin comparison model), we performed stratified Cox regression or conditional logistic regression, depending on the outcome. We assigned a unique identifier to cousins born to MZ twin parents and stratified our analysis based on this identifier. We corrected the between- and within-twin-family $p$-values for multiple testing ($N = 9 * 3$) with the Benjamini–Hochberg false discovery rate [17].

We included all cousins, regardless of whether they were differentially exposed to parental psychiatric conditions, as the covariate information contributed to the likelihood [18]. However, only cousins who were discordant for both the exposure and outcome contribute to the statistical power. To account for the dependence of individuals, we estimated

PLOS Medicine

robust standard errors by clustering on the MZ parent identifier. We assessed the between- and within-twin-family effects both without and with adjusting for covariates. Original analyses, using data up to December 2013, were planned verbally between May and July 2023, with analyses carried out between September 2023 and September 2024. In response to reviewers' comments, we re-ran the analyses with data up to December 2020 (see S2 Appendix for study proposal).

### Sensitivity analyses

We conducted five sensitivity analyses. First, because the effect of parental psychiatric conditions on the offspring might be assumed to be weaker when children have grown into adults, we analyzed a subsample where we restricted the exposure to parental psychiatric conditions occurring prior to age 18 of the offspring (i.e., we excluded parents whose first psychiatric condition were diagnosed after the child had turned 18 years old). Second, acknowledging that lifetime parental diagnoses include conditions predating childbirth—which may not fully align with developmental theories emphasizing the importance of the childrearing environment—we analyzed a subsample where we restricted the exposure to parental psychiatric conditions occurring after childbirth. Third, to account for potential cohort effects (e.g., changes in diagnostic practices, healthcare access) and to ensure that cousins had roughly similarly long follow-up periods, we excluded cousins who were born more than 8 years apart [19]. We selected an 8-year cousin age gap because it provides sufficient temporal separation to minimize potential cohort effects while retaining adequate sample sizes for statistical power and aligns with our previous sibling comparison study [19]. Fourth, to examine whether the effects varied by follow-up age, we divided the study population into three birth cohorts: 1970–1980, 1980–1990, and 1990–2000, so that individuals were aged 41–51, 31–41, and 21–31 years, respectively, at the end of follow-up. Fifth, to ensure that consolidating different psychiatric outcomes does not obscure differences among them, we also analyzed the associations between parental psychiatric conditions and each specific offspring outcome.

Data management was done using SAS version 9.4. Data analyses were performed using R version 4.4.3 with the package survival [20].

## Results

We identified 15,603 unique individuals born to 7,742 MZ twin parents. Of the 15,603 individuals, 2,821 (1,392 parents), 1,791 (867 parents), and 1,180 (573 parents) had an MZ twin parent with any psychiatric condition, any internalizing, or any externalizing condition, respectively. Table 1 summarizes the demographics for both offspring and parent-specific variables from the 15,603 individuals. As only cousins who were discordant for both the exposure and outcome were informative for the within-twin-family estimates, we also reported this effective sample size in Table 2. The sample sizes of both exposure- and outcome-discordant cousins (cousin clusters) ranged from 577 (127) for externalizing behaviors to 2,801 (650) for internalizing conditions when examining the association with any parental psychiatric condition; from 390 (85) for externalizing behaviors to 1,996 (462) for internalizing conditions when examining associations with parental internalizing conditions; and from 321 (63) for suicide to 1,176 (265) for internalizing conditions when examining associations with parental externalizing conditions.

### Intergenerational associations between any parental psychiatric condition and offspring outcomes

In the covariate-adjusted between-family model, children with a parent who had any psychiatric condition exhibited significantly higher probabilities for all outcomes, compared to those whose parents had no such history, with HRs ranging from 1.51 (95% CI [1.41, 1.62]; $p < 0.001$) for offspring internalizing conditions to 1.83 (95% CI [1.59, 2.11]; $p < 0.001$) for offspring psychotic conditions (Fig 2 and Table C in S1 Appendix). Similarly, the OR of 1.46 (95% CI [1.32, 1.62]; $p < 0.001$) for offspring poor school performance and 1.47 (95% CI [1.32, 1.63]; $p < 0.001$) for offspring long-term unemployment. All associations remained significant after correcting for multiple testing.

**Table 1. Description of offspring- and parent-specific characteristics from the 15,603 offspring born 1970–2000.**

| | Total | Twin parent with any psychiatric condition | | Twin parent with any internalizing condition | | Twin parent with any externalizing condition | |
|---|---|---|---|---|---|---|---|
| | | No | Yes | No | Yes | No | Yes |
| **Offspring-specific characteristics N (%)** | | | | | | | |
| N | 15,603 | 12,782 | 2,821 | 13,812 | 1,791 | 14,423 | 1,180 |
| **Sex** | | | | | | | |
| Male | 8,096 (51.89) | 6,670 (52.18) | 1,426 (50.55) | 7,183 (52.01) | 913 (50.98) | 7,499 (51.99) | 597 (50.59) |
| Female | 7,507 (48.11) | 6,112 (47.82) | 1,395 (49.45) | 6,629 (47.99) | 878 (49.02) | 6,924 (48.01) | 583 (49.41) |
| **Birth year** | | | | | | | |
| 1970–1980 | 5,599 (35.88) | 4,583 (35.86) | 1,016 (36.02) | 5,033 (36.44) | 566 (31.60) | 5,207 (36.10) | 392 (33.22) |
| 1980–1990 | 5,440 (34.87) | 4,437 (34.71) | 1,003 (35.55) | 4,798 (34.74) | 642 (35.85) | 4,996 (34.64) | 444 (37.63) |
| 1990–2000 | 4,564 (29.25) | 3,762 (29.43) | 802 (28.43) | 3,981 (28.82) | 583 (32.55) | 4,220 (29.26) | 344 (29.15) |
| **Paternal age at childbearing, Mean (SD)** | 30.89 (5.53) | 30.94 (5.45) | 30.68 (5.86) | 30.91 (5.49) | 30.71 (5.78) | 30.92 (5.47) | 30.57 (6.17) |
| **Maternal age at childbearing, Mean (SD)** | 28.27 (4.86) | 28.34 (4.82) | 27.91 (5.03) | 28.31 (4.84) | 27.89 (5.04) | 28.33 (4.82) | 27.46 (5.26) |
| **Parent-specific characteristics N[a](%)** | 7,742 | 6,350 | 1,392 | 6,875 | 867 | 7,169 | 573 |
| **Spouse's psychiatric conditions** | | | | | | | |
| No | 6,356 (82.10) | 5,308 (83.59) | 1,048 (75.29) | 5,717 (83.16) | 639 (73.70) | 5,943 (82.90) | 413 (72.08) |
| Yes | 1,386 (17.90) | 1,042 (16.41) | 344 (24.71) | 1,158 (16.84) | 228 (26.30) | 1,226 (17.10) | 160 (27.92) |
| **Highest educational level** | | | | | | | |
| N[b] | 8,404 | 6,800 | 1,604 | 7,391 | 1,013 | 7,705 | 699 |
| Compulsory school (≤9 years) | 522 (6.21) | 413 (6.07) | 109 (6.80) | 457 (6.18) | 65 (6.42) | 465 (6.04) | 57 (8.15) |
| Upper secondary school | 3,986 (47.43) | 3,161 (46.49) | 825 (51.43) | 3,481 (47.10) | 505 (49.85) | 3,598 (46.70) | 388 (55.51) |
| University | 3,896 (46.36) | 3,226 (47.44) | 670 (41.77) | 3,453 (46.72) | 443 (43.73) | 3,642 (47.27) | 254 (36.34) |
| **Birth year (mother)** | | | | | | | |
| N[c] | 8,012 | 6,537 | 1,475 | 7,099 | 913 | 7,365 | 647 |
| Before 1950 | 2,481 (30.97) | 2,032 (31.08) | 449 (30.44) | 2,241 (31.57) | 240 (26.29) | 2,305 (31.30) | 176 (27.20) |
| 1950–1960 | 2,817 (35.16) | 2,311 (35.35) | 506 (34.31) | 2,513 (35.40) | 304 (33.30) | 2,579 (35.02) | 238 (36.79) |
| 1960–1970 | 2,234 (27.88) | 1,824 (27.90) | 410 (27.80) | 1,947 (27.43) | 287 (31.43) | 2,062 (28.00) | 172 (26.58) |
| After 1970 | 480 (5.99) | 370 (5.66) | 110 (7.46) | 398 (5.61) | 82 (8.98) | 419 (5.69) | 61 (9.43) |
| **Birth year (father)** | | | | | | | |
| N[d] | 8,119 | 65,991 | 1,520 | 7,152 | 967 | 7,496 | 623 |
| Before 1950 | 3,346 (41.21) | 2,729 (41.35) | 617 (40.59) | 2,994 (41.86) | 352 (36.40) | 3,108 (41.46) | 238 (38.20) |
| 1950–1960 | 2,680 (33.01) | 2,186 (33.13) | 494 (32.50) | 2,359 (32.98) | 321 (33.20) | 2,471 (32.96) | 209 (33.55) |
| 1960–1970 | 1,797 (22.13) | 1,462 (22.15) | 335 (22.04) | 1,561 (21.83) | 236 (24.41) | 1,657 (22.11) | 140 (22.47) |
| After 1970 | 296 (3.65) | 222 (3.36) | 74 (4.87) | 238 (3.33) | 58 (6.00) | 260 (3.47) | 36 (5.78) |

[a]The total number corresponds to parents who are the monozygotic twins = 7,742.

[b]The total number corresponds to the parent pair = 8,404.

[c]The total number corresponds to the mother = 8,012.

[d]The total number corresponds to the father = 8,119.

However, when comparing differentially exposed cousins whose parents were MZ twins (i.e., within-twin-family models), all estimates attenuated and several became statistically nonsignificant. Nevertheless, even after correcting for multiple testing, the associations of any parental psychiatric condition with any offspring psychiatric conditions (HR = 1.28, 95% CI [1.13, 1.44]; $p < 0.001$) and offspring internalizing conditions (HR = 1.25, 95% CI [1.14, 1.38]; $p < 0.001$) remained statistically significant (Fig 2 and Table C in S1 Appendix).

**Table 2. Sample size of the cousins (i.e., children of monozygotic twins) who were discordant for both exposure and outcome.**

| | Twin parent exposure | | |
| --- | --- | --- | --- |
| | Any parental psychiatric condition | Parental internalizing condition | Parental externalizing condition |
| **Offspring outcomes** | | | |
| Any psychiatric condition | 2,253 (505) | 1,693 (380) | 1,022 (223) |
| Psychotic conditions | 1,056 (234) | 840 (185) | 499 (105) |
| Neurodevelopmental conditions | 952 (211) | 804 (177) | 466 (97) |
| Internalizing conditions | 2,801 (650) | 1996 (462) | 1,176 (265) |
| Substance use disorders | 914 (196) | 650 (143) | 536 (109) |
| Externalizing behaviors | 577 (127) | 390 (85) | 397 (85) |
| Suicide | 683 (151) | 488 (110) | 321 (63) |
| Poor school performance | 1,702 (389) | 1,239 (281) | 818 (185) |
| Long-term unemployment | 1874 (430) | 1,348 (304) | 848 (187) |
| Discordant monozygotic twins[a] | 3,486 (855) | 2,464 (596) | 1,470 (352) |

Note: The sample size of cousin cluster was in parentheses.

[a]Tthe sample size of the cousins who were discordant for parental exposure.

## Intergenerational associations between parental internalizing conditions and offspring outcomes

In the covariate-adjusted between-family models, children with a parent who had any internalizing condition exhibited significantly higher probabilities for all outcomes, compared to those whose parents had no such history, with HRs ranging from 1.36 (95% CI [1.07, 1.72]; $p=0.012$) for offspring externalizing behaviors to 2.03 (95% CI [1.73, 2.38]; $p<0.001$) for offspring psychotic conditions and OR of 1.47 (95% CI [1.30, 1.66]; $p<0.001$) for offspring long-term unemployment and 1.33 (95% CI [1.17, 1.52]; $p<0.001$) for offspring poor school performance. All associations remained significant after correcting for multiple testing.

However, when comparing differentially exposed cousins whose parents were MZ twins (i.e., within-twin-family models), all estimates attenuated. Nevertheless, even after adjusting for multiple testing, the associations of parental internalizing conditions with any offspring psychiatric conditions (HR = 1.26, 95% CI [1.09, 1.45]; $p=0.002$), internalizing conditions (HR = 1.17, 95% CI [1.04, 1.31]; $p=0.009$), and poor school performance (OR = 1.35, 95% CI [1.07, 1.71]; $p=0.010$) remained statistically significant (Fig 2 and Table C in S1 Appendix).

## Intergenerational associations between parental externalizing conditions and offspring outcomes

In the covariate-adjusted between-family models, children with parents who had any externalizing condition exhibited significantly higher probabilities for all outcomes, compared to those whose parents had no such history, with HRs ranging from 1.34 (95% CI [1.21, 1.49]; $p<0.001$) for offspring internalizing conditions to 2.53 (95% CI [1.96, 3.26]; $p<0.001$) for offspring externalizing behaviors and OR of 1.87 (95% CI [1.63, 2.14]; $p<0.001$) for offspring long-term unemployment and 1.73 (95% CI [1.49, 2.01]; $p<0.001$) for offspring poor school performance. All associations remained significant after correcting for multiple testing.

However, when comparing differentially exposed cousins whose parents were MZ twins, all estimates attenuated. Nevertheless, even after correcting for multiple testing, the associations of any parental externalizing condition with any offspring psychiatric (HR = 1.27, 95% CI [1.08, 1.51]; $p=0.005$) and any neurodevelopmental conditions (HR = 1.45, 95% CI [1.09, 1.93]; $p=0.010$) remained statistically significant (Fig 2 and Table C in S1 Appendix).

## Sensitivity analyses results

Similar to the main analysis, the within-twin-family associations with the broad spectra appeared robust across sensitivity specifications, but a few additional associations emerged (even after correcting for multiple testing; Tables D and E

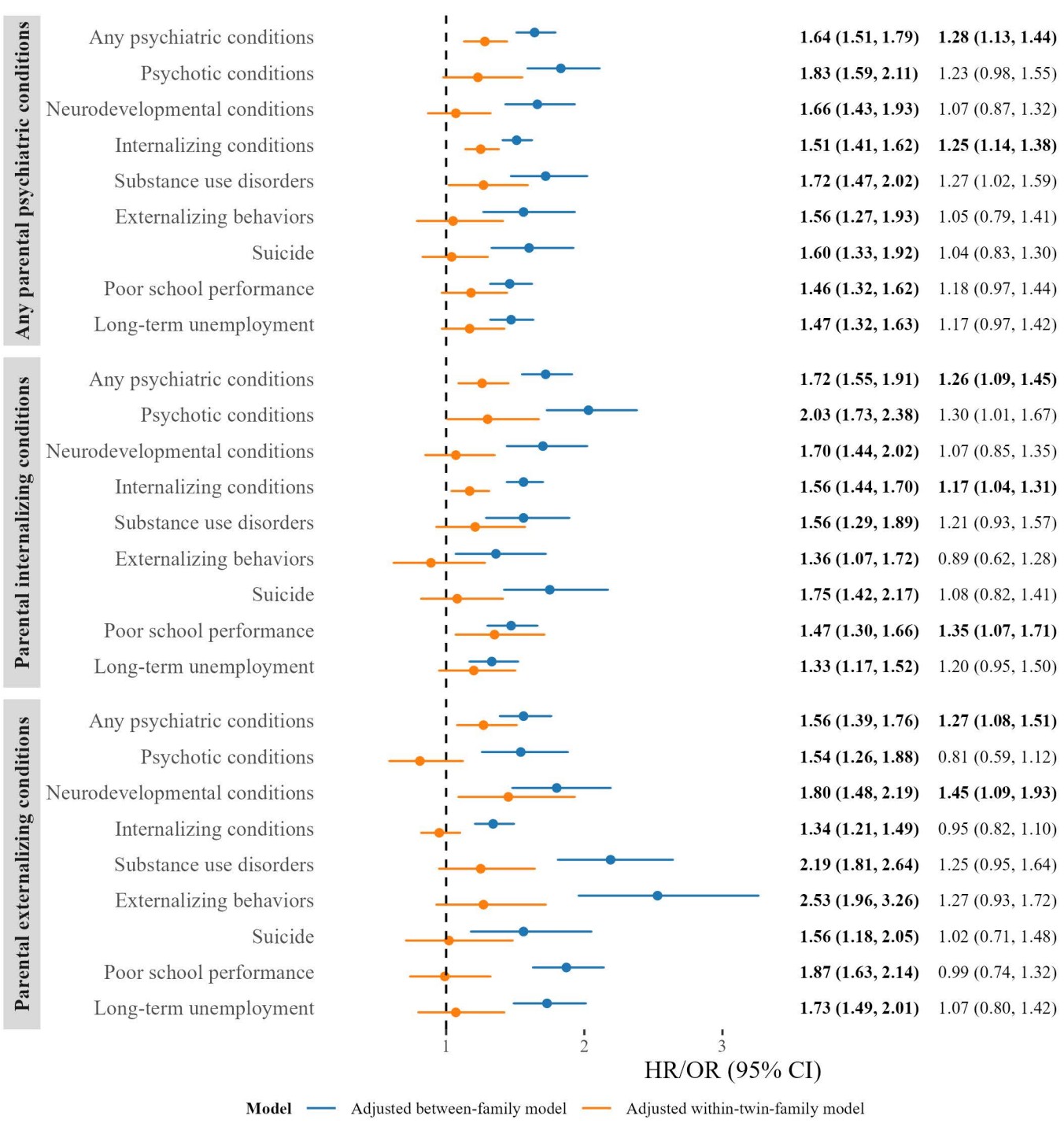

**Fig 2. Association between parental psychiatric conditions and offspring outcomes.** Note: Adjusted covariates include the highest parental educational level, any psychiatric condition of the nontwin parent, maternal and paternal age at childbirth, offspring birth year, and sex. The estimator is the odds ratio for poor school performance and long-term unemployment. Estimators shown in bold meet the criteria for Benjamini–Hochberg False Discovery Rate statistical significance (was conducted separately for between-family model [*N*=9 * 3] and within-twin-family model [*N*=9 * 3]). *P* values were reported in Table C in S1 Appendix.

in S1 Appendix). First, when restricting the analysis to parents whose first psychiatric conditions were diagnosed prior to their child reaching 18 years of age, the results were similar to the main analyses, with the exception that significant associations were additionally observed between any parental psychiatric condition and offspring neurodevelopmental conditions, and between parental internalizing conditions and offspring neurodevelopmental conditions. Second, when restricting the analysis to parents whose first psychiatric conditions were diagnosed after childbirth, the results were similar to the main analyses, with the exception that significant associations were additionally observed between any parental psychiatric condition and offspring substance use disorders, poor school performance, and between parental externalizing conditions and offspring substance use disorders. Third, when restricting the age difference between cousins to 8 years or less, the results remained similar to the main analyses, with the exception that significant associations were additionally observed between any parental psychiatric condition and offspring long-term unemployment, and between parental internalizing conditions and offspring long-term unemployment. Fourth, when dividing the study population into three birth cohorts, most associations were nonsignificant due to wider CIs, with the exception that significant associations were additionally observed between parental internalizing conditions and offspring suicide in the 1970–1980 cohort. Fifth, when analyzing associations with each specific offspring outcome, we observed statistically significant associations of any parental psychiatric condition with offspring prescription of anxiolytics and antidepressants, of parental internalizing conditions with offspring prescription of anxiolytics, and of parental externalizing conditions with offspring ADHD, prescription of stimulants, and PTSD.

## Discussion

Similar to past studies [1–4], we observed that children with parents who had internalizing or externalizing conditions exhibited higher probabilities for psychiatric, behavioral, and adverse psychosocial outcomes. However, when comparing cousins whose parents were identical twins and where one twin parent had a psychiatric condition and the other twin parent did not, the associations attenuated and 20 out of 27 became statistically nonsignificant. In other words, even though one cousin was exposed to a twin parent with a psychiatric condition and the other cousin was not, both cousins had similar probabilities of suffering many outcomes. This suggests that familial (genetic or environmental) factors that influence both generations account for most of the intergenerational transmission of psychiatric and behavioral problems.

Nevertheless, significant (but attenuated) associations remained between parental psychiatric conditions and any offspring psychiatric conditions in the covariate-adjusted within-twin-family model. This association could be attributable to either nonshared factors, or to a causal effect. If the latter, that might indicate that being raised by a parent with a psychiatric vulnerability shapes the rearing environment (e.g., suboptimal parenting practices, household chaos, instability, financial stress) that in turn contributes to the development of psychiatric vulnerability in the offspring. This would highlight the importance of environmental interventions, such as supporting parental mental health, for breaking the intergenerational cycle of psychiatric risk. In support of this, RCT indicates that comprehensive interventions, such as targeting not only the parental psychiatric condition(s) but also aiming to support parenting, offspring functioning, or even the whole family, seem more efficacious than narrower interventions at breaking the intergenerational transmission of mental health problems [21,22].

Although our study did not decompose the familial factors into that which can be attributed to genetic versus shared environmental factors, we speculate that genetic factors likely feature centrally in the intergenerational transmission because of the high heritability of most psychiatric disorders and because the unmeasured environmental factors shared by cousins that are adjusted for in the COT design are often relatively small [5]. Indeed, a recent systematic review that included studies that decomposed familial factors into genetic and shared environmental subparts also concluded that genetic transmission plays an important role [23]. In addition, recent family-based polygenic risk scores studies based on parents and/or offspring consistently demonstrate that direct genetic transmission constitutes the primary mechanism in intergenerational transmission of mental health problems [24–29].

Consistent with our significant within-twin-family association between parental internalizing conditions and offspring internalizing conditions, children of twin/sibling studies have also found that parental internalizing conditions remain associated with offspring internalizing conditions after adjusting for unmeasured familial factors [10,30–34]. However, in contrast to our null findings, some children of twin/sibling studies have also reported that parental internalizing conditions were associated with offspring externalizing behaviors [30,32,33], ADHD [35], and conduct disturbance after controlling for unmeasured familial factors [31]. These diverging results could have three explanations. First, all of the above studies combined the children of DZ twins or even children of full- and half-siblings and used structural equation models to decompose the observed association between the parental and child phenotype into genetic and environmental sources. This method requires more assumptions than our design (e.g., it assumes that the shared environment is the same for twin and sibling pairs). Second, most of the above studies did not control for the psychiatric status of the spouse, which could lead to bias by factors not shared by cousins. Third, the above survey studies were based on parent- or self-reported symptoms or diagnoses, which are potentially subject to recall or rater bias [36,37]. In contrast, the psychiatric conditions in our study were assigned by psychiatrists, such that recall and rater biases were likely to be minimal.

Our mostly null within-twin-family associations of parental externalizing conditions with offspring outcomes is consistent with a systematic review of COT studies and a COT study using the Vietnam Era Twin Registry [5,38], which also found that the associations between parental substance use disorders and offspring substance use disorders, externalizing problems, or internalizing problems were attributable to unmeasured familial factors. Aside from replicating their findings in a larger sample, we extended the evidence into adulthood. Although one COT study observed a significant association between parental alcohol/drug dependence and offspring conduct problems [13], this study did not compare cousins born to MZ twins; instead, they compared offspring with affected parents to unrelated offspring with unaffected parents but with affected parent-MZ-co-twin (uncle/aunt), such that the observed effect might still be attributable to residual confounding. In contrast to the statistically significant association between parental externalizing and offspring neurodevelopmental conditions we observed in this study, a systematic review of COT studies found no direct environmental effect of parental alcoholism on offspring ADHD [5].

Aside from psychiatric outcomes, our study adds to the existing literature by also examining the risk of suicide and adverse psychosocial outcomes in the offspring. Although observational studies have shown that parental psychiatric conditions are associated with an increased risk of these outcomes in offspring [39–44], only a few have used genetically informative designs to rule out familial factors. In line with our null within-twin-family results, a Swedish registry-based adoption study showed that an adoptive mother's psychiatric hospitalization did not increase the risk of suicide attempts among adoptees [45]. Similarly, a Norwegian register-based adoption study observed a null association between adoptive parents' internalizing disorders and school performance among adoptees [46]. However, we observed a statistically significant association between parental internalizing conditions and offspring (poor) school performance. To our knowledge, no studies to date have examined the association between parental psychiatric conditions and offspring unemployment after controlling for unmeasured familial factors. Further studies using genetically informative designs or other natural experiments are warranted to explore whether our within-twin-family estimate replicates.

Although we used the to-date largest (that we are aware of) sample of children-of-identical twins from nationwide and longitudinal registers in Sweden and followed them into adulthood, several limitations are noted. First, importantly, despite controlling for unmeasured familial factors shared by cousins, the COT designs cannot isolate unmeasured factors not shared by cousins. Although we controlled for potential observed nonshared factors, such as psychiatric diagnoses of the nontwin parent, unobserved residual confounding could remain. On a related note, although we treated psychiatric diagnoses of the nontwin parents, highest parental educational level, and maternal and paternal age at childbirth as confounders in this study, they might also be mediators such that one should not (necessarily) control for them. Nevertheless, these covariates contributed only minimally to the results, such that the conclusion remained consistent regardless of whether they might best be conceptualized as confounders versus mediators. Second, as is typical of within-cluster

analyses (e.g., fixed effects regression), the imprecision was slightly greater for the within-family estimates compared to the between-family estimates because only cousins discordant for parental psychiatric conditions were informative for the within-cluster associations. On a related note, measurement error (or unreliability) plays a larger role for the within- compared to between-family estimates; we tried to minimize this by summing the exposures according to classical test theory [47,48], but it could still contribute to the reduction in the within-family estimates. Third, the use of only children of Swedish MZ twins may limit the generalizability of our findings to the general nontwin population or to other countries. Fourth, we included only clinical diagnoses assigned in specialist care (i.e., we lacked information on individuals treated exclusively in primary care or those who did not seek treatment), so the associations are likely generalizable to cases identified in specialist care only. Fifth, if there were carryover effects (i.e., if psychopathology in one adult twin parent were to influence the outcomes in the offspring of their cotwin), this would also reduce the within-family estimates [49]. Sixth, although a study found a high positive predictive value (PPV) of 84% for psychiatric diagnoses [50], the accuracy varies across conditions. Specifically, the corresponding PPV was 81% for bipolar disorder [51], 72% for OCD [52], and 92% for tic disorders [52]. This variation in diagnostic accuracy may introduce different misclassifications across psychiatric conditions, which we assume is nondifferential with respect to exposure and outcome, a scenario that would typically underestimate true associations [53]. Seventh, although our study represents the largest children-of-identical-twins sample to date, statistical power remained limited for specific clustered outcomes, preventing us from drawing strong conclusions regarding these outcomes. Eighth, although we controlled for psychiatric diagnosis of the nontwin parent, residual confounding from assortative mating might persist in both the between- and within-family models, which could potentially lead to overestimated associations. However, partner choice might function not as a confounder (as typically assumed in assortative mating scenarios) but as a mediator. If so, controlling for partner characteristics could constitute over-correction and lead to underestimated associations. Ninth, our use of lifetime parental diagnoses did not ensure temporal precedence, as some conditions may have been diagnosed before the birth or after the outcome of their child. While our sensitivity analyses restricting parental diagnoses occurring before the offspring's adolescence (before age 18) and to the postchildbirth period demonstrated robust effects, future studies could employ stricter temporal definitions for exposure. Tenth, although associations remained significant for certain specific outcomes in the within-family model (e.g., parental internalizing conditions were associated with offspring poor school performance), these associations were not statistically significant across all the sensitivity analyses. Therefore, it is important to explore if this association replicates in other samples or with other quasi-experimental designs.

The associations of any parental, internalizing, and externalizing conditions with offspring psychiatric conditions, externalizing behaviors, suicide, poor school performance, and long-term unemployment attenuated and many became statistically nonsignificant in the within-twin-family model, suggesting that shared familial factors account for most of the observed associations. Nevertheless, statistically significant within-family associations remained between broad parental psychiatric spectra and any offspring psychiatric condition. If these associations are not attributable to unmeasured non-shared factors (i.e., attributable to direct causal effects), then treating parental psychiatric conditions might reduce the risk of psychiatric vulnerability in offspring.

## Supporting information

**S1 Checklist. STrengthening the Reporting of OBservational studies in Epidemiology (STROBE) Statement—checklist of items that should be included in reports of observational studies, available at** https://www.strobe-statement.org/**, licenced under CC BY 4.0.**
(DOC)

**S1 Appendix. Table A.** Description of registries and variables extracted. **Table B.** The ICD/ATC code, classified convictions for violent crimes, and the cut-off age for exposures and outcomes. **Table C.** Association between parental

psychiatric conditions and offspring outcomes, HR/OR (95% CI). **Table D.** Association between parental psychiatric conditions and offspring outcomes in the adjusted within-twin-family model, HR/OR (95% CI), sensitivity analyses. **Table E.** Association between parental psychiatric conditions and each specific offspring outcome in the adjusted within-twin-family model, HR (95% CI).
(DOCX)

**S2 Appendix.  Project proposal.**
(PDF)

## Author contributions

**Conceptualization:** Erik Pettersson.

**Data curation:** Henrik Larsson, Brian M. D'Onofrio, Ralf Kuja-Halkola, Zheng Chang, Isabell Brikell, Paul Lichtenstein, Erik Pettersson.

**Formal analysis:** Mengping Zhou.

**Funding acquisition:** Erik Pettersson.

**Methodology:** Mengping Zhou, Erik Pettersson.

**Project administration:** Paul Lichtenstein.

**Resources:** Erik Pettersson.

**Supervision:** Mikael Landén, Paul Lichtenstein, Erik Pettersson.

**Writing – original draft:** Mengping Zhou.

**Writing – review & editing:** Mengping Zhou, Henrik Larsson, Brian M. D'Onofrio, Mikael Landén, Ralf Kuja-Halkola, Zheng Chang, Isabell Brikell, Paul Lichtenstein, Erik Pettersson.

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
