## [Editor Report · Decision Letter 0]

4 Feb 2025

Dear Dr Zhou,

Thank you for submitting your manuscript entitled "Intergenerational Transmission between Parental Psychiatric Conditions and Offspring Psychiatric, Behavioral, and Psychosocial Outcomes: A Swedish Population-Based Children-of-Monozygotic Twins Study" for consideration by PLOS Medicine.

Your manuscript has now been evaluated by the PLOS Medicine editorial staff as well as by an academic editor with relevant expertise and I am writing to let you know that we would like to send your submission out for external peer review.

Please re-submit your manuscript within two working days, i.e. by Feb 06 2025 11:59PM.

Kind regards,

Suzanne

Suzanne De Bruijn, PhD

Associate Editor

PLOS Medicine

---

## [Decision Letter · Decision Letter 1]

29 May 2025

Dear Dr Zhou,

Many thanks for submitting your manuscript "Intergenerational Transmission between Parental Psychiatric Conditions and Offspring Psychiatric, Behavioral, and Psychosocial Outcomes: A Swedish Population-Based Children-of-Monozygotic Twins Study" (PMEDICINE-D-25-00393R1) to PLOS Medicine. The paper has been reviewed by three subject experts and a statistician; their comments are included below and can also be accessed here: [LINK]

As you will see, the reviewers found the study and methodology interesting, but they raised a number of questions and request a number of clarifications, including for example, the use of lifetime diagnosis of the parent which might allow for the diagnosis to precede the offsprings’ birth. After discussing the paper with the editorial team and an academic editor with relevant expertise, I'm pleased to invite you to revise the paper in response to the reviewers' comments. Please note that we plan to send the revised paper to some or all of the original reviewers, and we cannot provide any guarantees at this stage regarding publication.

We ask that you submit your revision by June 19th. However, if this deadline is not feasible, please contact me by email, and we can discuss a suitable alternative.

Don't hesitate to contact the handling editor with any questions (sbruijn@plos.org).

Kind regards,

Heather

Heather Van Epps, PhD

Consulting Editor

[on behalf of]

Suzanne De Bruijn, PhD

Associate Editor

PLOS Medicine

sbruijn@plos.org

Comments from the reviewers:

Reviewer #1 (statistics):

This is an interesting study that examined the intergenerational transmission between parental psychiatric conditions and their offspring's outcomes. It could have strong implication on how the transmission should be tackled, and the use of a children-of-monozygotic twins design is excellent in handling how familial / genetic factors being controlled.

The study is well conducted and the reporting is with sufficient details. The statistical methods were appropriate. The findings are important and interesting. I have several minor comments:

1. I'm a bit reserved about the claim that the non-significant findings means no associations, i.e. the claim 'fully attributable to unmeasured familial and genetic factors' in the conclusion. The estimated 95% CIs are quite wide (0.45 (95% CI: 0.20, 1.02) to 1.57 (95% CI: 0.83, 2.96)) and still compatible with strong associations.

2. If indeed authors want to provide stronger support to this claim, a post-hoc power analysis may be able to show detectable effect size from this study.

3. The end of follow-up was quite dated - over 10 years ago. Would there be any notable diagnostic differences that makes the findings less applicable to today? A short discussion may be useful.

Reviewer #2:

The present study examines intergenerational transmission of mental disorders using children of twins approach in Swedish registry data. Results demonstrate that although parental mental disorders were related with increased risk of offspring mental disorders, the association were diluted in the within-twin-family analyses.

Although the present study approach is novel and the results are very interesting, there are number of issues that require further clarification.

(1) Although study is based on large-scale register, the statistical power (based on table 2) seems rather limited in many of the analyses. Most of the data is currently available until to the end of 2013. Would be possible to update the data up to 2023, for example, to increase statistical power?

(2) Using life-time diagnosis as the main exposure seems like not the best possible solution, and rational for this is currently missing. The first sensitivity analyses partially address this issue, but ideally parental mental disorders would be diagnosed before the follow-up for offspring mental disorders begins to avoid reverse causality.

(3) The interpretation of the findings warrants further discussion, as the between-family and within-family models target different causal estimands (the first estimates family-level causal effect and the second family level causal export among exposure-discordant cousins). Thus, the findings from these two models are not directly comparable without additional assumptions.

(4) Given the high lifetime incidence of mental disorders (up to 80% in Kessing et al. JAMA Psychiatry 2023), the prevalence of parental mental disorders in the study cohorts seems rather low. What could explain this and how does it influence the present findings?

(5) In addition to examining parental externalizing and internalizing disorders, I would like to see an analysis where any parental mental disorders would be used as an exposure.

(6) Discussion related to the validity of the mental disorders diagnoses is currently missing.

Reviewer #3:

Zhou and colleagues investigate the intergenerational transmission of psychiatric conditions using high-quality Swedish national registry data and a sophisticated children-of-MZ-twins (CoMZ) design. Strengths of this study include the use of clinician-assigned psychiatric diagnoses, a broad range of offspring psychiatric and psychosocial outcomes, and thoughtful sensitivity analyses. The manuscript is well-written, the analyses are rigorous, and the Discussion is nuanced and appropriately contextualized. Below, I provide specific suggestions to further strengthen the manuscript.

1. The claim that intergenerational transmission is "fully attributable" to familial factors may be overstated. While the attenuation of associations in the within-twin-family models indeed reflect familial factors, the substantial reduction in effective sample size (e.g., some outcomes had fewer than 100 discordant cousin clusters) could limit the power to detect small but potentially meaningful environmental effects. I would suggest that authors interpret these non-significant findings more cautiously. Additionally, the Discussion would benefit from further consideration of the limited statistical power inherent in the CoMZ design, the possibility of residual confounding, and the fact that some associations (e.g., between parental internalizing conditions and offspring school performance or unemployment) became statistically significant in the age-restricted sensitivity analysis.

2. While transdiagnostic approaches are valuable, the rationale for grouping specific psychiatric conditions into internalizing and externalizing domains should be more clearly explained. For example, the inclusion of PTSD, OCD, and eating disorders under internalizing conditions warrants justification. Additionally, while parental psychiatric conditions were grouped into two broad domains (internalizing vs. externalizing), offspring outcomes were analyzed across nine distinct clusters. Although the authors mention this was done to increase power, a clearer theoretical or methodological rationale for the asymmetry between exposure and outcome domains would strengthen the justification.

3. Although Tables 1 and 2 provide detailed sample size information, the effective sample sizes used in the between-family and within-twin-family analyses are not explicitly stated in the Results section. Given that the interpretation of null findings depends on statistical power, I recommend briefly reiterating the analytic sample sizes, particularly the number of discordant cousin pairs contributing to the within-family models (e.g., range of N across outcomes), either at the beginning of the Results or within each relevant subsection.

4. The finding that parental internalizing conditions were significantly associated with poor school performance and long-term unemployment in the sensitivity analysis restricting cousin age difference (≤8 years) is important. I suggest more explicitly exploring potential explanations in the Discussion.

5. While partner psychiatric diagnosis is included as a covariate, it may not fully account for intergenerational effects arising from assortative mating or genetic nurture. These mechanisms could contribute to residual confounding in both between- and within-family models. A brief discussion of how these complexities might influence the current findings would be informative and help contextualize the strengths and limitations of the CoMZ design.

6. Recent family-based genetic studies increasingly use polygenic risk scores (PRS) from both parents and offspring to investigate intergenerational psychiatric risk. I recommend briefly situating the current findings within this growing body of literature and explaining how the CoMZ design complements these genome-based approaches.

Other minor comments:

- Please add a clear explanation of the distinction between between-family and within-twin-family analyses.

- Please clarify how missing data (e.g., covariates, psychiatric outcomes) were handled, for instead complete case analysis, imputation, or otherwise.

- Justify the rationale for selecting an 8-year cousin age gap threshold in the sensitivity analysis. Citing a relevant reference or providing examples (e.g., changes in diagnostic practices or healthcare access) would improve interpretability.

- If the sample permits, consider testing or discussing potential sex differences in intergenerational transmission patterns.

- Clarify the number of tests for FDR correction: (N = 18; 2 exposure categories × 9 offspring outcomes).

Reviewer #4:

Thank you for inviting me to review this manuscript, which examines associations between parent psychiatric conditions and offspring outcomes. The study uses administrative data from Sweden and uses a clever "children-of-twins" design (essentially comparing cousins) to rule out genetic confounding. I think this is a very well-written manuscript, addressing an important question using sophisticated methods. The methods and findings are described very clearly (which is worth highlighting, given that the CoT design is quite complicated). I only had a few concerns:

- Offspring where between age 14 and 44 years old. It would be helpful to know if effects varied by age - 14 seems quite young for a psychiatric disorder diagnosed (and therefore captured in the registry data) as well as for some of the other outcomes, so it would be helpful to know if the analyses of younger outcome ages may have under-estimated effects. I understand that the analysis is underpowered as is, but even just descriptively presenting this comparison would be helpful.

- If I understand correctly, the authors use lifetime diagnoses in the parent, which means that the parent could have been diagnosed before the child was born. This seems a bit problematic given that most theories of the intergenerational transmission of mental health problems would probably argue that child exposure to parental mental health problems is what matters (rather than, say, having a parent who was diagnosed with a mental health problem once before their child was born). If possible, it would be good to do sensitivity analyses taking this into account. This should also be pointed out as a limitation (if I'm correct).

- Why did the analysis focus on externalising and internalising, rather than also taking into account psychotic-like conditions (as was done for the offspring, and as was mention for a previous study in the Introduction)? I imagine this may have been due to power issues - it would be helpful to clarify.

- It would be helpful to present (in the appendix) estimates of whether being a child of an MZ twin is predictive of any of the outcomes, just to pre-empt concerns that MZ twin offspring might be different from general population offspring.

- In the first section of the Results, please present the effective sample size in the text, or at least give an indication (e.g. report the range if the n varies across outcomes) - some of these n's drop to quite small numbers so I think it could be a bit misleading to only report those impressively-looking large numbers in the text.

- The Discussion focused on the null effects, but if I follow correctly, then within-family associations were significant for parental internalising problems and offspring educational attainment and unemployment, which is worth highlighting.

- The use of diagnoses is a strength, but it's also a limitation, because they capture only a subset of people with a mental health problem. This would be worth pointing out in the limitations section.

---

(Note: not all will apply to your paper, but please check each item carefully and include your responses in your point-by-point response, noting N/A where approrpriate)

*We ask every co-author listed on the manuscript to fill in a contributing author statement, making sure to declare all competing interests. If any of the co-authors have not filled in the statement, we will remind them to do so when the paper is revised. If all statements are not completed in a timely fashion this could hold up the re-review process. If new competing interests are declared later in the revision process, this may also hold up the submission. Should there be a problem getting one of your co-authors to fill in a statement we will be in contact. Please do not add or remove authors without first discussing this with the handling editor. You can see our competing interests policy here: http://journals.plos.org/plosmedicine/s/competing-interests.

*Please upload any figures associated with your paper as individual TIF or EPS files with 300dpi resolution at resubmission; please read our figure guidelines for more information on our requirements: http://journals.plos.org/plosmedicine/s/figures. While revising your submission, please upload your figure files to the PACE digital diagnostic tool, https://pacev2.apexcovantage.com/. PACE helps ensure that figures meet PLOS requirements. To use PACE, you must first register as a user. Then, login and navigate to the UPLOAD tab, where you will find detailed instructions on how to use the tool. If you encounter any issues or have any questions when using PACE, please email us at PLOSMedicine@plos.org.

*Please ensure that the paper adheres to the PLOS Data Availability Policy (see http://journals.plos.org/plosmedicine/s/data-availability), which requires that all data underlying the study's findings be provided in a repository or as Supporting Information. For data residing with a third party, authors are required to provide instructions with contact information (web or email address) for obtaining the data. Please note that a study author cannot be the contact person for the data. PLOS journals do not allow statements supported by "data not shown" or "unpublished results." For such statements, authors must provide supporting data or cite public sources that include it.

*We expect all researchers with submissions to PLOS in which author-generated code underpins the findings in the manuscript to make all author-generated code available without restrictions upon publication of the work. In cases where code is central to the manuscript, we may require the code to be made available as a condition of publication. Authors are responsible for ensuring that the code is reusable and well documented. Please make any custom code available, either as part of your data deposition or as a supplementary file. Please add a sentence to your data availability statement regarding any code used in the study, e.g. "The code used in the analysis is available from Github [URL] and archived in Zenodo [DOI link]" Please review our guidelines at https://journals.plos.org/plosmedicine/s/materials-software-and-code-sharing and ensure that your code is shared in a way that follows best practice and facilitates reproducibility and reuse. Because Github depositions can be readily changed or deleted, we encourage you to make a permanent DOI'd copy (e.g. in Zenodo) and provide the URL.

*Please ensure that the study is reported according to the STROBE guideline and include the completed STROBE checklist as Supporting Information. When completing the checklist, please use section and paragraph numbers, rather than page numbers. Please add the following statement, or similar, to the Methods: "This study is reported as per STROBE guideline (S1 Checklist)."

*Abstract: Please structure your abstract using the PLOS Medicine headings (Background, Methods and Findings, Conclusions). Please combine the Methods and Findings sections into one section.

*At this stage, we ask that you include a short, non-technical Author Summary of your research to make findings accessible to a wide audience that includes both scientists and non-scientists. The Author Summary should immediately follow the Abstract in your revised manuscript. This text is subject to editorial change and should be distinct from the scientific abstract. Ideally each sub-heading should contain 2-3 single sentence, concise bullet points containing the most salient points from your study. In the final bullet point of 'What Do These Findings Mean?', please include the main limitations of the study in non-technical language. Please see our author guidelines for more information: https://journals.plos.org/plosmedicine/s/revising-your-manuscript#loc-author-summary.

*Please express the main results with 95% CIs as well as p values. When reporting p values please report as p<0.001 and where higher as the exact p value p=0.002, for example. Throughout, suggest reporting statistical information as follows to improve clarity for the reader "22% (95% CI [13%,28%]; p</=)". Please be sure to define all numerical values at first use.

*Please include page numbers and line numbers in the manuscript file. Use continuous line numbers (do not restart the numbering on each page).

*Please cite the reference numbers in square brackets. Citations should precede punctuation.

FIGURES AND TABLES

*Please provide titles and legends for all figures and tables (including those in Supporting Information files).

*Please define all abbreviations used in each figure/table (including those in Supporting Information files).

*Please consider avoiding the use of red and green in order to make your figure more accessible to those with color blindness.

SUPPLEMENTARY MATERIAL

*Please note that supplementary material will be posted as supplied by the authors. Therefore, please amend it according to the relevant comments outlined here.

*Please cite your Supporting Information as outlined here: https://journals.plos.org/plosmedicine/s/supporting-information

REFERENCES

*PLOS uses the numbered citation (citation-sequence) method and first six authors, et al.

*Please ensure that journal name abbreviations match those found in the National Center for Biotechnology Information (NCBI) databases (http://www.ncbi.nlm.nih.gov/nlmcatalog/journals), and are appropriately formatted and capitalised.

*Where website addresses are cited, please include the complete URL and specify the date of access (e.g. [accessed: 12/06/2023]).

*Please also see https://journals.plos.org/plosmedicine/s/submission-guidelines#loc-references for further details on reference formatting.

OBSERVATIONAL STUDIES

*Abstract: Please include the study design, population and setting, number of participants, years during which the study took place (enrollment and follow up), length of follow up, and main outcome measures.

*Please ensure that the study is reported according to the STROBE (or appropriate STOBE extension) guideline (available from: https://www.equator-network.org/reporting-guidelines/strobe) and include the completed STROBE (or STROBE extension) checklist as Supporting Information. Please add the following statement, or similar, to the Methods: "This study is reported as per the Strengthening the Reporting of Observational Studies in Epidemiology (STROBE) guideline (S1 Checklist)." When completing the checklist, please use section and paragraph numbers, rather than page numbers.

*For all observational studies, in the manuscript text, please indicate: (1) the specific hypotheses you intended to test, (2) the analytical methods by which you planned to test them, (3) the analyses you actually performed, and (4) when reported analyses differ from those that were planned, transparent explanations for differences that affect the reliability of the study's results. If a reported analysis was performed based on an interesting but unanticipated pattern in the data, please be clear that the analysis was data driven.

*Please state in the Methods section whether the study had a prospective protocol or analysis plan. If a prospective analysis plan (from your funding proposal, IRB or other ethics committee submission, study protocol, or other planning document written before analyzing the data) was used in designing the study, please include the relevant document(s) with your revised manuscript as a Supporting Information file to be published alongside your study and cite it in the Methods section. A legend for this file should be included at the end of your manuscript. If no such document exists, please make sure that the Methods section transparently describes when analyses were planned, and when/why any data-driven changes to analyses took place. Changes in the analysis, including those made in response to peer review comments, should be identified as such in the Methods section of the paper, with rationale.

---

## [Decision Letter · Decision Letter 2]

19 Sep 2025

Dear Dr. Zhou,

Thank you very much for re-submitting your manuscript "Intergenerational Transmission between Parental Psychiatric Conditions and Offspring Psychiatric, Behavioral, and Psychosocial Outcomes: A Swedish Population-Based Children-of-Monozygotic Twins Study" (PMEDICINE-D-25-00393R2) for review by PLOS Medicine.

I have discussed the paper with my colleagues and the academic editor and it was also seen again by all four reviewers. I am pleased to say that provided the remaining editorial and production issues are dealt with we are planning to accept the paper for publication in the journal.

[LINK]

We look forward to receiving the revised manuscript by Sep 26 2025 11:59PM.   

Sincerely,

Suzanne De Bruijn, PhD

Associate Editor 

PLOS Medicine

plosmedicine.org

Requests from Editors:

GENERAL EDITORIAL REQUESTS

* Please change your title to: “Association between parental Psychiatric Conditions and Offspring Psychiatric, Behavioral, and Psychosocial Outcomes: A Swedish Population-Based Children-of3 Monozygotic Twins Study".

* Please confirm that your abstract complies with our requirements, including format (three sections: Background, Methods and Findings, and Conclusions) and providing all the information relevant to this study type https://journals.plos.org/plosmedicine/s/submission-guidelines#loc-abstract

* Please ensure that all abbreviations are defined at first use throughout the text.

* Please confirm that all numbers presented in the abstract are present and identical to numbers presented in the main manuscript text.

GENERAL

* Please remove the 'conclusions' subheading from the discussion. Please also remove any other subheadings from the discussion.

* Statistical reporting: Please revise throughout the manuscript, including tables and figures.

- Please report statistical information as follows to improve clarity for the reader ""22% (95% CI [13,28]; p</=)"".

- Please separate upper and lower bounds with commas instead of hyphens as the latter can be confused with reporting of negative values.

- Please repeat statistical definitions (HR, CI etc.) for each set of parentheses.

* In the abstract, please include the important dependent variables that are adjusted for in the analyses.

* In the author summary, in the final bullet point of 'What Do These Findings Mean?', please include the main limitations of the study in non-technical language.

* In the ethics statement, please include the IRB approval number.

FIGURES AND TABLES

* When a p value is given, please specify the statistical test used to determine it in the legend.

*in the legends, please specify all covariates that are adjusted for. They may be all present, but as it states ‘include’ this is not quite clear.

OBSERVATIONAL, COHORT, CROSS-SECTIONAL, AND CASE CONTROL STUDIES

* Did your study have a prospective protocol or analysis plan? Please state this (either way) early in the Methods section.

c) In either case, changes in the analysis-- including those made in response to peer review comments-- should be identified as such in the Methods section of the paper, with rationale."

* For all observational studies, in the manuscript text, please indicate: (1) the specific hypotheses you intended to test, (2) the analytical methods by which you planned to test them, (3) the analyses you actually performed, and (4) when reported analyses differ from those that were planned, transparent explanations for differences that affect the reliability of the study's results. If a reported analysis was performed based on an interesting but unanticipated pattern in the data, please be clear that the analysis was data-driven.

* Your study is observational and therefore causality cannot be inferred. Please remove language that implies causality and refer to associations instead.

Comments from Reviewers:

Reviewer #1: With the updated data and revised conclusion the paper reads very robust and relevant. I have no further comments.

Reviewer #2: Authors have addressed all the concerns that I raised. I have no further comments and would like to congratulate authors for the excellent work they have done.

Reviewer #3: The authors have revised the manuscript according to my previous comments. All of my suggestions have been adequately addressed, and I have no further comments.

Reviewer #4: The authors have been very responsive to all the reviewer comments and the manuscript has further improved as a result, from an already very strong baseline. No further concerns.

[LINK]

---

## [Editor Report · Decision Letter 3]

7 Oct 2025

Dear Dr. Zhou,

Thank you very much for re-submitting your manuscript "Association between Parental Psychiatric Conditions and Offspring Psychiatric, Behavioral, and Psychosocial Outcomes: A Swedish Population-Based Children-of-Monozygotic Twins Study" (PMEDICINE-D-25-00393R3) for review by PLOS Medicine.

As mentioned in our email correspondence, I have 3 remaining requests.

1) include the clarification regarding the existence of an analysis plan in the methods, and include the study proposal as a supplement.

2) Thank you for including the limitations in your author summary, but this bullet point is quite long. Could you change this into 2 bullet points? A) ‘the children-of-MZ-twins design does not control for unmeasured factors, meaning residual confounding factors could still be present’ and b) ‘This study is based on clinical diagnoses assigned in specialist care, and therefore likely not generalizable to cases treated exclusively in primary care, or to individuals who did not seek treatment’.

3) Could you shorten your abstract? it is currently over the maximum of 500 words allowed. I would suggest removing several of the explanation within parentheses (see below).

We look forward to receiving the revised manuscript by Oct 14 2025 11:59PM.   

Sincerely,

Suzanne De Bruijn, PhD

Associate Editor 

PLOS Medicine

plosmedicine.org

---

## [Editor Report · Decision Letter 4]

8 Oct 2025

Dear Dr Zhou, 

On behalf of my colleagues and the Academic Editor, Louisa Degenhardt, I am pleased to inform you that we have agreed to publish your manuscript "Association between Parental Psychiatric Conditions and Offspring Psychiatric, Behavioral, and Psychosocial Outcomes: A Swedish Population-Based Children-of-Monozygotic Twins Study" (PMEDICINE-D-25-00393R4) in PLOS Medicine.

PRESS

Sincerely, 

Suzanne De Bruijn, PhD 

Associate Editor 

PLOS Medicine